# Crosstalk of Brain and Bone—Clinical Observations and Their Molecular Bases

**DOI:** 10.3390/ijms21144946

**Published:** 2020-07-13

**Authors:** Ellen Otto, Paul-Richard Knapstein, Denise Jahn, Jessika Appelt, Karl-Heinz Frosch, Serafeim Tsitsilonis, Johannes Keller

**Affiliations:** 1Julius Wolff Institute for Biomechanics and Musculoskeletal Regeneration, Charité-Universitätsmedizin Berlin, 13353 Berlin, Germany; ellen.otto@charite.de (E.O.); denise.jahn@charite.de (D.J.); jessika.appelt@charite.de (J.A.); serafeim.tsitsilonis@charite.de (S.T.); 2Clinic of Trauma, Hand and Reconstructive Surgery, University Medical Center Hamburg-Eppendorf, 20246 Hamburg, Germany; p.knapstein@uke.de (P.-R.K.); k.frosch@uke.de (K.-H.F.)

**Keywords:** brain, bone, interaction, clinical and experimental studies, molecular signaling

## Abstract

As brain and bone disorders represent major health issues worldwide, substantial clinical investigations demonstrated a bidirectional crosstalk on several levels, mechanistically linking both apparently unrelated organs. While multiple stress, mood and neurodegenerative brain disorders are associated with osteoporosis, rare genetic skeletal diseases display impaired brain development and function. Along with brain and bone pathologies, particularly trauma events highlight the strong interaction of both organs. This review summarizes clinical and experimental observations reported for the crosstalk of brain and bone, followed by a detailed overview of their molecular bases. While brain-derived molecules affecting bone include central regulators, transmitters of the sympathetic, parasympathetic and sensory nervous system, bone-derived mediators altering brain function are released from bone cells and the bone marrow. Although the main pathways of the brain-bone crosstalk remain ‘efferent’, signaling from brain to bone, this review emphasizes the emergence of bone as a crucial ‘afferent’ regulator of cerebral development, function and pathophysiology. Therefore, unraveling the physiological and pathological bases of brain-bone interactions revealed promising pharmacologic targets and novel treatment strategies promoting concurrent brain and bone recovery.

## 1. Introduction

While the brain is regarded as the principal coordinator of body homeostasis by regulating organ activity and their crosstalk, bone features hematopoietic, endocrine metabolic and storage functions along with its predominant mechanical role. Although brain and bone seem apparently unrelated, exceptional clinical and experimental evidence propose a bilateral dependence of both organs [1,2]. The effect of brain on bone homeostasis and regeneration, transmitted via the ‘efferent’ nervous system, is well established [3,4,5] whereas the understanding of the ‘afferent’ effect of bone on brain function and development is still evolving [6,7,8]. Therefore, multiple stress, mood and neurodegenerative brain pathologies were previously correlated with bone loss, while only a limited number of genetic skeletal diseases was associated with the modulation of brain development and function. Trauma in particular was discovered to effect brain and bone concurrently.

In this review, we first recapitulate clinical observations and confirming experimental studies, demonstrating brain-bone interconnection. Thereafter, we provide a detailed overview of the molecular bases regarding these bilateral interactions. Based on this mechanistic understanding, we review promising therapeutic targets for disorders affecting brain and bone.

## 2. Clinical Observations

In the clinical setting, the ‘efferent’ effect of brain on bone remodeling is eventually reflected in bone gain or loss, the latter being the most common. Loss of bone density, strength and microarchitecture leads to the degenerative skeletal disorder osteoporosis, which occurs when the physiological homeostasis is disturbed and bone resorption of osteoclasts exceeds bone formation by osteoblasts. Osteoporosis represents the most common cause for fractures in the aging population, posing a major clinical issue and a significant socioeconomic burden [9]. Bone mineral density (BMD) is assessed through dual-energy x-ray absorptiometry (DXA) in which osteoporosis is defined at T-score −2.5 or less standard deviations below the average of young and healthy adults [10]. Osteoporosis has been associated with a great variety of brain dysfunctions such as epilepsy [11], schizophrenia [12], shift work [13], post-traumatic stress disorder [14], depression [15] as well as major neurodegenerative diseases including stroke [16], Alzheimer’s [17] and Parkinson’s disease [18]. Interestingly, trauma to the central nervous system such as traumatic brain [19] and spinal cord injury positively affect bone regeneration (for a detailed review please see Reference [20]). Finally, complex regional pain syndrome following trauma or surgery was reported to affect brain and bone concurrently [21].

The ‘afferent’ effect of bone on brain function is more difficult to elucidate and far less understood. None the less, rare genetic skeletal disorders are associated with changes of brain activity, potentially caused by interconnecting molecular mechanisms. Furthermore, peripheral bone injury was discovered to negatively modulate or even exacerbate traumatic brain injury [22]. In combination with growing molecular knowledge on bone-derived mediators, these clinical observations provided a first understanding of the skeletal capacity to modulate brain development and function.

### 2.1. Efferent ‘Brain-Bone’

Neuropsychological dysfunction, caused by shift work, post-traumatic stress and depression as well as the major neurodegenerative pathologies, are associated with bone loss and elevated fracture risk (Table 1) [23]. Considering the impact of brain dysfunction on bone however, direct interactions have to be distinguished from secondary effects following cognitive impairment and long hospitalization which commonly lead to reduced physical activity and altered mechanical loading [24]. Malnourishment, lifestyle behavior and psychotropic medication represent additional factors causing bone loss, the latter particularly implied for epilepsy [11] and schizophrenia [12].

Growing evidence provides a strong correlation between shift work and osteoporosis [13,25,26], as working nights are accompanied by numerous endocrinological changes [27] such as reduced levels of melatonin [28,29,30] and elevated levels of stress-induced cortisol [31]. These alterations were associated with an increase in body mass index [31], elevated risk for cardiovascular diseases [32], diabetes [33,34] and low BMD. As bone turnover markers were identified to mirror diurnal oscillations [35], bone remodeling is considered to follow central and peripheral circadian control [36,37]. Therefore, chronic inadequate sleep [38] and a disturbance in the expression of circadian clock genes have been observed to alter the skeletal phenotype [37,39,40].

Post-traumatic stress disorder represents another chronic psychological stress condition associated with bone loss, which poses an elevated risk for osteoporosis in civilian [14] as well as military patients [41]. Although malnutrition could not be excluded, the negative effect on bone mass was proposed to be mediated by elevated serum levels of proinflammatory cytokines such as tumor necrosis factor α (TNFα), interleukin 1 (IL1) and interleukin 6 (IL6) [42] known to stimulate bone resorption [43], as well as through hormones released in response to chronic psychological stress [44]. Stress signaling is predominantly mediated through the hypothalamic-pituitary-adrenal (HPA) axis, whereby pituitary-released adrenocorticotrophic hormone stimulates glucocorticoid synthesis in the adrenal cortex. Along with their important homeostatic, metabolic and immunologic functions, glucocorticoids were shown to directly inhibit osteoblastogenesis [45], which results in reduced bone mass and higher fracture risk [46]. Experimental evidence reported an additional negative effect of glucocorticoids on endochondral ossification in the growth plate, constraining longitudinal and appositional bone growth in adolescent mice [47]. Chronic psychosocial stress was also shown to modulate the immune response through β-adrenoreceptor signaling, resulting in impaired fracture healing [48].

Major depressive disorder (MDD) refers to a psychiatric condition also associated with low BMD [49] and higher fracture risk [50,51,52]. Apart from reduced physical activity and psychotropic medication, bone loss of patients suffering from MDD was proposed to result from inflammatory, metabolic and HPA axis dysregulations [53] with high levels of cortisol and catecholamines [54,55] as well as lower levels of steroids [56,57]. MDD is thought to impair bone and brain in a bidirectional manner, as low BMD and elevated fracture risk potentially result in pain conditions, further deteriorating depression [15]. MDD itself was additionally identified to negatively affect fracture healing through a direct inhibition of osteoblast differentiation [57].

Patients suffering from major neurodegenerative diseases including stroke, Alzheimer’s and Parkinson’s disease are commonly diagnosed with osteoporosis, resulting in high morbidity and mortality [58,59,60]. First, ischemic or hemorrhagic stroke leads to cell death and breakdown of the blood-brain barrier (BBB) [61]. Poststroke fracture is a common complication [62,63], which poses a substantial disadvantage for stroke recovery [63] as a result of immobilization, elevated bone resorption, hypercalcemia and hypovitaminosis D [64]. Interestingly, within the first few days following an acute stroke event, patients already display elevated serum concentrations of bone turnover markers including osteoprotegerin, sclerostin [65], dickkopf-related protein 1 [66] and osteopontin [67,68], suggesting direct correlation of stroke and bone loss. However, hypoxic conditions following stroke were shown to activate angiogenesis and osteogenetic precursor cells, resulting in heterotopic ossification in various parts of the body [69]. While vascular endothelial growth factor (VEGF) signaling is essential for adequate callus formation, hypoxic induction of VEGF further promotes brain edema [70,71]. Thus, experimental fracture, induced shortly before stroke, increased neuroinflammation and further exacerbated ischemic cerebral injury causing substantial secondary damage in a murine stroke-model [72].

Second, Alzheimer’s disease (AD), a chronic neurodegenerative disease, is also closely associated with osteoporosis and an increased fracture risk [73,74,75]. Patients with less brain atrophy show better bone quality [76], indicating central mechanisms of AD contributing to bone loss [7]. Further, patients suffering from AD or mild cognitive impairments display higher levels of osteopontin [77], which correlates with cognitive decline [78] and reduced BMD [79], while AD progression is linked to serum levels of the bone turnover markers osteopontin, osteocalcin and sclerostin [8]. In AD, an accumulation of extracellular amyloid-β (Aβ) plaques and intracellular tau inclusions causing cell degeneration has been observed [80]. In turn, Aβ was identified to increase osteoclast activation and bone resorption [81]. Thus, targeting of Aβ plaques may evolve as a promising therapeutic approach to prevent cognitive decline and bone loss in patients with AD [82].

Third, in line with stroke and AD, patients with Parkinson’s disease (PD) additionally display reduced BMD [18]. While the pathogenesis of PD includes oxidative stress, disturbances of iron metabolism [83] and aggregation of α-synuclein protein [84,85], the associated bone loss already occurs during early stages of disease development. As a result, the majority of PD patients are prone not only to neurological impairment and postural imbalance but also an increased fracture risk [18,86]. Along with correlating vitamin D deficiency, reduced body weight [87] and female gender [88], recent evidence propose a direct effect of PD on bone through degeneration of dopaminergic neurons, resulting in accelerated osteoclastogenesis and suppressed bone formation [89].

### 2.2. Afferent ‘Bone-Brain’

Although most clinical observations primarily highlight the impact of neurologic disorders on bone integrity, a limited number of genetic bone pathologies are accompanied by structural and cognitive brain impairment, pointing towards an ‘afferent’ bone-brain effect. In this regard, cleidocranial dysplasia (CCD) represents an autosomal dominant skeletal disorder caused by the haploinsufficiency of *RUNX2* (also called *CBFA1*), which is a key transcription factor of osteoblast differentiation. CCD is characterized by skeletal anomalies including brachycephalic skull, collarbones partly or completely missing, midfacial hypoplasia and delayed tooth eruption [90]. Some patients with CCD additionally suffer from a developmental delay of the brain or late-onset progressive cognitive decline [91]. This might potentially be explained by dysfunctional osteoblasts and an insufficient secretion of the osteoblast-derived hormone osteocalcin [92], which was shown to exert neuroprotective effects [6].

Similar to CCD, Coffin-Lowry syndrome (CLS) refers to a genetic skeletal disorder associated with brain malfunction. CLS represents an X-linked disease, caused by loss-of-function mutations in the gene *RPS6KA3* encoding for the growth-factor-regulated protein kinase RSK2, which phosphorylates activating transcription factor 4 (ATF4, also called CREB-2) [93]. Although clinical manifestation is highly heterogeneous, CLS patients show profound growth retardation with facial, hand and skeletal malformations as well as serious psychomotor impairments [94]. While the inactivation of RSK2 was associated with severe impaired spatial learning and long-term spatial memory deficit [95], long-term potentiation requires the RSK2 substrate ATF4 [96]. The skeletal malformations are also presumed to be caused by the lack of ATF4, which was shown to regulate osteoblast differentiation and function [94,97] as well as to stimulate osteocalcin expression [98]. Therefore, brain and bone dysfunction of patients suffering from CLS may be explained by altered ATF4 activity [1].

Other skeletal disorders, such as the *SOST* gene mutations sclerosteosis and van Buchem disease, are associated with raised intracranial pressure and cranial nerve entrapment [99,100], while hereditary multiple exostoses (HME) [101] correlates with symptoms of autism [102] and frontotemporal dementia [103]. Although multiple genetic diseases with concurrent skeletal and mental deficits (selection see Appendix A) show individually altered brain and bone dysfunction, further evidence of bidirectional molecular interaction is warranted.

### 2.3. Trauma Affecting Brain and Bone

Clinical studies involving physically injured patients revealed a strong crosslink of brain and bone. In response to general trauma or surgical injury, complex regional pain syndrome (CRPS) was observed to effect the nervous system and bone concurrently [21]. Although not directly affecting brain function, CRPS is characterized by autonomic, sensory and motoric abnormalities with clinical features of neurogenic inflammation, maladaptive neuroplasticity and nociceptive sensitization accompanied by sensory impairments, potentially leading to anxiety and depression [104,105,106]. In bone, CRPS results in loss of BMD and increased periarticular bone turnover with osteoprotegerin proposed as a potential biomarker [107].

Different to general trauma, the external physical insult to the head, causing an alteration of brain function among other brain pathologies, is termed traumatic brain injury (TBI) [108]. Depending on force severity, TBI results in temporary to permanent neurologic dysfunctions as well as a disruption of the circadian rhythm, behavioral and cognitive impairments, with generally increased mortality and morbidity [109,110,111]. During primary TBI, the meninges are commonly damaged causing cerebral hematoma, edema, ischemia and necrosis, leading to a disruption of the BBB [110,112]. Subsequent metabolic disturbance, apoptosis, oxidative stress and neuroinflammation are defined as secondary injury, potentially lasting for weeks [113] with wide-ranging systemic effects on the immune system and other organ function including bone [114]. Clinical TBI studies reported an ‘efferent’ effect on the intact bone, showing patients suffering from isolated TBI to exhibit an elevated fracture risk and reduced BMD [19,115,116]. Although immobilization represents a contributing factor, experimental studies confirmed the negative effect of TBI on bone quality and mineral density without changes in movement [117,118]. This alteration in bone metabolism is stated to be caused by the elevated BBB permeability, peripheral inflammatory response as well as endocrine and sympathetic outflow modulation of the secondary injury [114,119]. Recent evidence implies that the inflammatory stress on bone and its marrow following TBI activates nuclear factor ‘kappa-light-chain-enhancer’ of activated B-cells (NF-κB), which in turn induces osteoclastic differentiation resulting in elevated bone resorption [120]. Along with the negative impact on bone metabolism, patients suffering from TBI were furthermore identified to frequently sustain heterotopic ossification (HO) [121,122]. Such musculoskeletal ectopic deposition of lamellar bone in non-osseous tissue is commonly acquired after neurological, soft or bone tissue trauma [119], especially following combined TBI and fracture or high severity injury [123]. Although it has been agreed that neurological HO, initiated by simultaneous central and peripheral nervous injury, represents endochondral ossification, the underlying molecular and cellular mechanisms still remain to be elucidated [122,124,125]. Recently established TBI models sustaining HO potentially provide the foundation for investigations of the pathogenesis while unraveling promising therapeutic targets [126,127].

In contrast to isolated brain injury, patients suffering from TBI with concomitant fracture however were identified to exhibit accelerated bone healing and enhanced callus formation [128,129,130,131,132,133,134]. Similar to the physiologic bone remodeling, bone healing represents a complex process, consisting of the pro- and anti-inflammatory phase, followed by the soft callus, hard callus and the remodeling phases [135,136]. Although continues attempts were made to unravel the underlying molecular bases of this positive ‘efferent’ TBI effect on bone regeneration [137], the exact mechanisms still remain to be elucidated. Clinical studies monitored the systemic regulation of trauma patients suffering from fracture and TBI, to identify potential osteogenic humoral factors [138]. Therefore, proliferation was significantly increased when treating osteoblasts with serum [139] or cerebrospinal fluid of TBI patients [140], pointing towards centrally released osteogenic factors entering the circulation following TBI [138]. For further investigation, experimental studies reproduced the phenomenon [141], reporting an increased bone volume, elevated mineral density and higher rates of gap bridging in mice with TBI and concomitant fracture [142]. As today, different theories have been postulated, considering a complex modulation of the inflammatory response, participating hormones, neuropeptides and neurotransmitters [114,119,137,143]. Most recent evidence proposed a non-humoral pathway with dominance of neuronal mechanisms and neuroinflammation [144]. Similar to TBI, isolated spinal cord injury (SCI) was associated with reduced BMD and osteoporosis [145], predominantly observed within the trabecular metaphysical-epiphyseal areas of the distal femur and proximal tibia [146,147,148,149]. Further, patients suffering from SCI also frequently show HO as well as accelerated and enhanced callus formation [20], which was reproduced in vitro utilizing serum of SCI patients [150,151].

Despite the positive effect of trauma on bone regeneration, the ‘afferent’ bone-brain interaction of peripheral injury such as fracture was discovered to negatively modulate and potentially further deteriorate TBI [22]. While a neurological impact of fracture healing is considered to be constrained by the BBB, trauma patients suffering from TBI and concomitant skeletal injury show higher functional deficits and mortality rates [151]. Clinical multitrauma studies identified systemic inflammatory changes [152,153,154] with the capacity to modulate the neuroinflammatory response following TBI [22]. Such alterations were strongly supported by experimental evidence, reporting concomitant fracture to exacerbate TBI, neuroinflammation [155] and further deteriorate cerebral edema, motor deficits and neurological recovery [156].

## 3. Molecular Bases of Brain-Bone Crosstalk

Clinical observations continuously unravel pathophysiological processes and interactions, which reveal potential therapeutic benefit and therefore commonly represent the bases for specific mechanistic research. As a result, the combination of clinical and experimental studies gave rise to a continuously improving understanding of the brain-bone crosstalk. Several mutations in the genes encoding the below referenced mediators were discovered in humans to affect development and metabolism of bone and brain, respectively, to various degrees (Appendix A). In this section, we provide an insight to molecules which directly mediate the signal transmission of these two organs.

### 3.1. Brain- and Nerve-Derived Mediators Affecting Bone Cell Function

#### 3.1.1. Central Regulation

The central nervous system (CNS) is classically known for its major role in coordinating the activity of all parts of the body through neuroendocrine signaling which is primarily funneled by the hypothalamus. Most of these regulatory mediators are either expressed by the hypothalamic nuclei and transported to the posterior pituitary gland or secreted to stimulate hormone release in the anterior pituitary gland. Notably, those hormones, including follicle-stimulating hormone, thyroid-stimulating hormone, prolactin, adrenocorticotrophic hormone, growth hormone, arginine vasopressin, oxytocin and pineal gland-derived melatonin were all discovered to represent potent regulators of skeletal integrity (Figure 1, Table 2) [4,170].

Growing evidence supports an estrogen-independent, direct effect of **follicular stimulating hormone** (FSH) on extragonadal tissue, particularly bone and fat [171,172,173,174]. Although this matter remains intensively discussed [175,176,177,178,179], osteoclasts and their precursor cells have been shown to express FSH receptors, allowing FSH to directly stimulate osteoclast formation, function and survival [173,174]. Bone resorption is additionally promoted indirectly following the upregulation of pro-resorptive cytokines in proportion to FSH receptor expression [179]. Especially during late perimenopause, when estrogen levels are still unaltered although ovarian failure is impending, a sharp increase of serum FSH levels was associated with an enhanced rate of bone loss and visceral adiposity onset, proposing FSH as a potential key player in osteoporosis and obesity in women across the menopausal transition [171,178]. Therefore, FSH-neutralizing antibodies were tested in pre-clinical experiments, which lead to increased bone mass and reduced body fat [171,180].

In line with FSH, **thyroid stimulating hormone** (TSH) was shown to directly affect bone remodeling through binding to the TSH receptor expressed in both osteoblast and osteoclast precursors, independent of thyroid T3 and T4 [181]. TSH was identified to negatively regulate osteoclastogenesis trough binding to TSH receptor directly but also indirectly by suppressing the synthesis of pro-osteoclastic signals [181,182] such as the cytokine TNFα, a critical mediator of the antiresorptive effects of TSH [183]. However, TSH was not only observed to reduce bone loss but also to restore bone mass, bone volume, microarchitecture and strength [184]. In osteoblasts, TSH induces the synthesis of noncanonical Wnt5a, resulting in osteoblastogenesis and stimulation of osteoprotegerin (OPG) synthesis, which in turn reduces bone resorption through an inhibition of receptor activator of nuclear factor κB ligand (RANKL) signaling [185]. However, TSH receptor activation was demonstrated to suppress osteoblast differentiation and the expression of collagen type 1 by impairing the Wnt pathway and decreasing VEGF concentrations [181]. Although these studies define TSH as a single and independent control molecule in bone formation as well as resorption, further investigation are required to unravel the underlying molecular pathophysiology and its therapeutic potential.

Lactotrophic cell-derived prolactin (PRL) is predominantly known for its pivotal role in lactation, mammary development and reproduction, additionally participating in bone homeostasis [186]. Interestingly, G protein-coupled PRL receptors are only expressed on osteoblasts but not osteoclasts [187], with a variable response based on PRL concentration levels. Although PRL is essential for bone growth and homeostasis [187], patients with pathological high PRL concentrations show increased bone resorption and suppressed bone formation activity, which eventually leads to osteoporosis [188]. On a mechanistic level, several studies observed PRL to reduce osteocalcin and alkaline phosphate activity [189], to decrease OPG expression [190,191] and to inhibit osteoblastic proliferation as well as bone mineralization [192]. Besides these direct effects, hyperprolactinemia is also proposed to negatively act on bone in an indirect manner, involving hypogonadism, hypercalcemia and an enhanced secretion of parathyroid hormone-related peptide (PTHrP) [188]. As PRL treatment however showed a positive effect on bone formation in infant rats [193] and a decreased RANKL/OPG ratio in human fetal osteoblasts [192], the effect of PRL on bone is proposed to be dependent on the developmental stage [194] and warrants further mechanistic understanding.

Adrenocorticotrophic hormone (ACTH), the key mediator of corticosteroid-release from the adrenal gland, represents an important co-regulator of immune responses, vascular tone, central metabolism and bone turnover [194,195]. Chronically elevated levels of glucocorticoids are a well-established cause for osteoporosis and presumbly osteonecrosis due to their inhibitory impact on bone-forming osteoblasts [47,196]. Zaidi et al. propose that ACTH in interaction with VEGF supports the prevention of glucocorticoid-induced osteonecrosis [170,194,195]. Additional studies described the expression of ACTH by monocytes/macrophages [197], which acts cortisol-independent on melanocortin 2 receptors (MC2R) expressed by osteoblastic cells [195]. Therefore, local and pituitary-derived ACTH features the capacity to regulate bone cells directly [170] in a concentration-dependent manner [195].

While most anterior pituitary gland-derived hormones show a divergent, rather negative effect on bone remodeling, growth hormone (GH) also known as somatotropin, is a peptide hormone crucial for human development, stimulating cell reproduction and regeneration as well as the regulation of longitudinal bone growth [198]. GH was shown to induce a net gain of bone mass [198], with in vivo studies demonstrating the capacity of GH to stimulate the proliferation of osteoblastic-cells [199]. Although the majority of experimental evidence implies GH to exert primarily insulin-like growth factor 1 (IGF-1)-dependent skeletal effects, the group of Zaidi et al. currently challenge this view by proposing the direct effect of GH on bone cells [170,194].

In contrast to all neurohormones previously discussed, arginine-vasopressin (AVP) commonly known as antidiuretic hormone, is a peptide hormone released from the posterior pituitary gland, which has also been reported to regulate bone metabolism [4]. In bone, AVP receptors (Avpr1α/2) are expressed by osteoblasts and osteoclasts, allowing AVP-binding to reduce osteoblastogenesis as well as to increase osteoclastogenesis [200]. As patients with chronic hyponatremia show severe osteoporosis and high fracture risk, their elevated AVP levels were suggested to mobilize sodium from skeletal cells which increases bone resorption [170,194]. Additionally, AVP was shown to promote the production of corticotropin-releasing hormone, which in turn facilitates the release of ACTH [201].

Oxytocin (OT), also released from the posterior pituitary gland, is known for its role in lactation, parturition [157,202], social behavior, as well as in energy and bone homeostasis [203,204]. Interestingly, OT and AVP were observed to interact with each other’s G protein-coupled receptors in order to control bone metabolism [203]. As opposed to AVP, OT shows a rather anabolic effect on bone [204], promoting osteoblastogenesis while moderately inhibiting osteoclast activity [205]. OT receptor expression was identified in osteoblasts [206] and osteoclasts [207] while bone marrow cells were shown to produce OT themselves, implying an autocrine/paracrine regulation of bone formation [204]. Notably, OT synthesis and OT receptor expression were positively regulated by estrogen signaling [208], explaining the observed sex differences in experimental and clinical studies [171].

Different to all pituitary-derived hormones, melatonin is primarily released from the pineal gland into the cerebrospinal fluid [158,209] and physiologically involved in the synchronization of circadian and seasonal rhythms including the sleep-wake cycle with further regulation of the blood pressure and temperature homeostasis [3]. In bone, both osteoblasts and osteoclasts were observed to express melatonin receptors [210]. As bone remodeling is commonly known to follow circadian rhythmic, melatonin was early discovered to support osteoblastic differentiation [211] and type 1 collagen synthesis [212]. As a result melatonin was identified to inhibit adipogenesis and promote osteogenesis [213], which was further confirmed by the observation of melatonin to induce the expression of bone morphogenetic proteins, alkaline phosphatases, osteocalcin, OPG and the suppression of RANKL [159]. While in vivo studies led to contradictory results [3], clinical observation showed a gain of BMD following melatonin supplementation in postmenopausal women with osteopenia [214], fueling continuous discussions on melatonin’s therapeutic potential [159,215,216].

Complementary to all these neurohormones involved in the central regulation of bone remodeling, further neuropeptides and -transmitters, the endocannabinoid system as well as the central clock (Figure 1) play an important role in bone homeostasis. While the control of energy homeostasis is generally divided among anabolic-active AgRP/NPY and catabolic-active POMC/CART neurons, their impact on bone metabolism appears to be more complex [4]. The multifunctional and orexigenic neuropeptide Y (NPY) is predominantly expressed in the arcuate nucleus (ARC) [217], regulating energy metabolism through induced food intake and increased fat storage [218]. In the peripheral nervous system (PNS), NPY is mainly expressed by neurons of the sympathetic nervous system were it plays a major role in bone metabolism through its receptors Y1 and Y2 [219,220]. When binding to Y1 expressed by osteoblasts [221], NPY inhibits bone formation, proliferation of mesenchymal stem cells and osteoprogenitors [222] and promotes energy storage in white adipose tissue [223], while binding to Y2 reduces bone mass and osteoblast activity through central modulation [3,220,221]. During chronic stress however, NPY was reported to feature protective effects on bone [224], which is potentially explained by a centrally-mediated decrease in sympathetic tone [221].

The neuropeptide agouti-related peptide (AgRP) is co-expressed with NPY. AgRP represents one of the most potent and long-lasting appetite stimulators, in addition to its modulatory impact on metabolism and energy expenditure [225,226]. Interestingly, it has been proposed that altered AgRP neuronal activities affect bone homeostasis independent of metabolic shifts or leptin signaling [227,228]. In this regard, enhanced AgRP neuronal activity was reported to lower sympathetic tone, which favors increased bone mass due to elevated osteoblast activity [229,230].

Cocaine amphetamine regulated transcript (CART), an anorexic neuropeptide precursor protein also involved in the regulation of food intake and energy expenditure [3,231,232,233,234], is released in the ventral tegmental area of the brain according to the serum levels of leptin [234]. While low hypothalamic CART expression was associated with increased bone resorption through higher levels of RANKL, elevated CART expression resulted in increased bone mass phenotype [217,233]. Although the pituitary gland and pancreatic islets additionally release CART into the system [234], no direct effect on osteoblast gene expression was observed [233].

Proopiomelanocortin (POMC) is a large precursor protein for multiple peptide hormones including ACTH, melanocyte-stimulating hormones (MSHs) and beta-endorphin [235]. The melanocortin peptides ACTH and α-, β- and γ-MSH bind with varying affinity to five known melanocortin receptors (MCRs) belonging to the group of G protein-coupled receptors [236]. While all POMC peptides feature immunomodulatory capacity attenuating inflammatory processes [236], melanocortins were particularly identified to show osteo- and chondro-protective effects [237,238,239,240]. Therefore, α-MSH is proposed to potentially delay the process of inflammatory and degenerative joint diseases [240]. Additionally, single nucleotide polymorphisms (SNPs) in the promoter region of the POMC gene were associated with low BMD [241]. Female mice lacking estrogen receptors in POMC neurons display an increase in cortical bone mass and mechanical strength [242], which implies a negative effect of estrogen on bone mass through POMC neurons.

Neuromedin U (NMU), a neuropeptide predominantly expressed in the pituitary and gastrointestinal tract [243], is involved in the regulation of smooth muscle contraction, blood pressure, feeding, energy homeostasis, nociception and stress response [244]. A central, leptin-dependent, regulation of bone remodeling by NMU was demonstrated in NMU-deficient mice which showed higher bone mass due to increased bone formation [245]. In rats, NMU was reported to regulate the corticotropin releasing hormone as a stress response in the pituitary gland [243,246]. In this regard, elevated concentration of both ACTH and corticosterone were observed following NMU injections [247]. Furthermore, NMU stimulates the release of vasopressin and enhances the secretion of steroids from rat adrenal cortex [247]. Taken together, these observations point out potential treatment indications of NMU receptor agonists, which could target bone loss diseases or stress-related disorders without inducing unwanted body weight gain [248,249].

The vasoactive intestinal peptide (VIP) is synthesized in various tissues including the pancreas, gastrointestinal tract as well as the hypothalamus and the autonomic nervous system. VIP is commonly associated with the sympathetic (SNS) as well as the parasympathetic nervous system (PSNS) [250], as ACh and VIP are often co-released from parasympathetic nerve fibers [221] in the periosteum and epiphysis of bone [251,252,253]. G protein-coupled VIP receptors are expressed in both osteoblasts [254] and osteoclasts [255]. VIP was reported to directly suppress receptor activator of nuclear factor κB (RANK) in osteoclasts and RANKL in osteoblasts, while it increases the expression of OPG [256,257] causing an anti-resorptive effect.

The central regulation of bone homeostasis is furthermore controlled by important neurotransmitters of the CNS, including the monoamine serotonin and dopamine as well as the excitatory neurotransmitter glutamate (Figure 1). In bone, different subtypes of G protein-coupled serotonin receptors are expressed by osteoblasts and osteoclasts [258]. Gut-derived serotonin was reported to decrease osteoblast proliferation in a low-density lipoprotein receptor-related protein 5 (*Lrp5*)-dependent manner [259], as well as to promote maturation of osteoclasts resulting in elevated bone resorption [260]. In fact, it has not been clarified whether *Lrp5* signaling via the serotonin or Wnt pathway is more crucial [261]. Overall, the impact of serotonin on bone metabolism is proposed to be origin-dependent, whereby peripherally produced serotonin inhibits and central serotonin enhances bone formation [262]. These findings are supported by clinical data showing an acceleration of postmenopausal bone loss and reduction in bone mass accrual following the use of selective serotonin reuptake inhibitors (SSRIs) [263,264,265].

Dopamine (DA), a member of the catecholamine family, is widely expressed in the CNS and some peripheral areas, which binds to five different dopamine receptors (DR_1_-DR_5_). Signaling of DR_1_ [266], DR_2_ [267], DR_3_ and DR_5_ [268] was evidenced to enhance osteoblastic proliferation and bone mineralization as well as to suppress osteoclastogenesis [269]. The inhibition of these receptors by atypical antipsychotics leads to bone loss [270,271]. Dopamine represents another important molecule of bone homeostasis which gains further importance in diseases with dopamine decline such as AD, PD, depression or schizophrenia, while it demonstrates that the clinical use of neuroleptics potentially affects bone mass [271].

The excitatory neurotransmitter glutamate (GLU) activates ionotropic and metabotropic receptors. All mature bone cells feature glutamate receptors, with N-methyl-D-aspartate (NMDA) receptor being the most commonly expressed by osteoblasts [272] and osteoclasts [273,274]. Glutamate inhibits osteoclast activity [275] while promoting osteoblast differentiation and function [276]. In addition, it was shown that not only bone is innervated densely by glutamate-containing nerve fibers [277] but osteoblasts themselves secret glutamate through exocytosis [278]. Intracellularly, the enzyme glutamine synthase converts active glutamate into inactive glutamine, thus regulating the concentration of glutamate. Interestingly, the basal expression of glutamine synthase itself was shown to be regulated positively by glucocorticoids and negatively by Wnt signaling or Vitamin D [279], establishing glutamine synthase as a key player in the regulation of osteoblastogenesis.

The central regulation of bone remodeling is further controlled by the **circadian clock** [3] as well as the endocannabinoid system [280]. As the majority of homeostatic and metabolic functions are under circadian control, bone represents no exception, with central and peripheral circadian rhythms controlling bone remodeling. The central pacemaker is located in the hypothalamic suprachiasmatic nucleus, continuously synchronized with the daily light-dark cycle. The heterodimer of pacemaker transcription factors *Bmal1* and *Clock* regulate the gene expression of downstream targets such as periods (*Per1–3*) and cryptochromes (*Cry1*, *Cry2*) [281]. A cyclic expression oscillation of these genes was discovered to regulate glucocorticoids by elevating the hypothalamic secretion of ACTH [282] as well as the sympathetic outflow [283,284], both associated with a negative effect on bone mass. Although continuously synchronized with the central pacemaker, the existence of peripheral clocks was revealed following the observation of time dependent gene expression within different cell types [36,285]. In bone, circadian expression oscillation was primarily reported for osteoblast activity [286,287,288] but also identified in osteoclasts and osteocytes [289,290] as well as mesenchymal stem cells [291]. A further mechanistic understanding of the circadian system and bone remodeling relationship is thus expected to identify novel therapeutic strategies to treat major bone diseases such as osteoporosis.

Cannabinoid signaling is transduced by two cannabinoid receptors (CB1 and CB2), with CB1 predominantly expressed in presynaptic neurons of the CNS and PNS [292,293] and CB2 in peripheral tissues [294]. CB2 receptors expressed in osteoblasts and osteoclasts [295,296,297] were identified to promote bone formation as well as to represses RANKL, thereby inhibiting bone resorption [298]. In clinical studies, genetic variants of the *CNR2* gene, encoding for CB2 receptors, were associated with low BMD and osteoporosis [299]. In contrast, centrally expressed CB1 receptors are considered to decrease the sympathetic outflow by modulating adrenergic signaling [221]. In this regard, CB1 receptors are proposed to transmit retrograde signals which inhibit the release of the sympathetic nervous system transmitter norepinephrine and thereby stimulate bone formation [300,301]. As a result, CB1/CB2 agonists and antagonists are under investigation for their therapeutic capacity in bone regeneration [302]. Additionally, the phytocannabinoid cannabidiol was identified to stimulate collagen crosslinks and stabilize callus formation by stimulating lysyl-hydroxylase activity in osteoblasts while respecting the BBB, which promotes cannabidiol as a treatment option for osteoporosis [302] and impaired fracture healing [303].

#### 3.1.2. SNS and PSNS

The autonomic nervous system integrates input from the internal and external microenvironment to ensure essential body function and homeostasis, also in adaption to stressors [221]. The autonomic nervous system is separated into the sympathetic (SNS) and parasympathetic nervous system (PSNS), which are both substantially involved in the regulation of bone homeostasis [3,4,5,221].

The SNS, commonly known to prepare for alert situations [221], signals through the neurotransmitter norepinephrine (NE) which activates G protein-coupled α- and β-adrenergic receptors at effector organs. Noradrenergic fibers were identified within the periosteum, branching into the bone marrow and mineralized bone alongside blood vessels [221]. Although the expression of subtypes from both α- and β-adrenergic receptors were detected in both osteoblasts and osteoclast [221], osteoblastic β-adrenergic receptors are considered as the major transducers of sympathetic signaling in bone remodeling [304]. The activation of β2-adrenergic receptors was identified to inhibit osteoblast function and further shown to induce the release of osteoblast-derived RANKL, thus promoting osteoclast formation [221,233].

As the SNS is considered a negative regulator of bone mass, the PSNS is proposed to counterbalance this impact. In contrast to the SNS, the PSNS emerges from cranial nerves as well as the sacral spinal cord, innervating the rostral and caudal part of the body, respectively. The PSNS promotes restive and digestive responses with a rather positive effect on bone remodeling, signaling through acetylcholine (ACh) at muscarinic and nicotinic cholinergic receptors [221]. Although parasympathetic nerve fibers were discovered within the bone microenvironment [305], specific density and pattern of the cholinergic bone innervation still remains to be determined [221]. The expression of both cholinergic receptor types was identified in osteoclasts and osteoblasts, with osteoclasts being the predominant target of cholinergic signaling inducing an overall inhibitory effect [221]. Notably, ACh inhibition [306] and central muscarinic receptor deletion studies [307] revealed a significant ACh regulation on CNS-mediated bone remodeling. Exclusively the deletion of muscarinic ACh receptor 3 (M3R), which is expressed in different areas of the brain, resulted in significant lower bone formation with elevated bone resorption, identical to the phenotype observed following adrenergic agonists treatment [308]. Therefore, cholinergic signaling is proposed to promote bone mass accrual through the central decrease of sympathetic activity [307].

Historically, the skeletal innervation of the autonomic nervous system was first identified histologically, while its crucial function for bone remodeling started with the discovery of leptin [221]. Leptin represents an adipocyte-derived hormone, encoded by the *Ob* (*Lep*) gene. In bone, leptin was discovered to modulate bone metabolism through hypothalamic signaling, leading to an increase in cortical bone and a decrease in trabecular bone [308,309]. Along with its direct impact on the hypothalamus, leptin was observed to bind to hypothalamic receptors, which regulate bone metabolism through the SNS [310,311]. The impact of leptin on bone mass is also related to the intake of energy and IGF-1 pathway [312]. Partially opposing the influence of leptin, the second adipocyte-derived hormone adiponectin was observed to signal in neurons of the locus coeruleus, which decreases the sympathetic tone and therefore leads to an increase in bone mass.

#### 3.1.3. Sensory Innervation

Along with the SNS and PSNS, the sensory nervous system (SeNS) represents the third arm of the autonomic nervous system involved in the regulation of skeletal homeostasis [4]. A network of sensory nerve fibers was identified in compartments of trabecular bone, with rich innervation throughout the periosteum [313,314]. The impact of sensory innervation on bone emerged from denervation studies using capsaicin to disrupt sensory neurons [3,5]. The loss of sensory nerves caused elevated bone resorption with no influence on bone formation, overall resulting in reduced BMD [315,316].

Sensory nerves express calcitonin gene-related peptide and substance P as their main neurotransmitters [221] which substantially regulate bone homeostasis (Figure 1). Calcitonin gene-related peptide (CGRP) represents a neuropeptide occurring as two major isoforms (α and β), both encoded by different genes while sharing similar biological activities [317]. CGRP, expressed in both the CNS and PNS, primarily functions as a potent vasodilator. In bone, CGRP-positive nerve fibers were reported in the periosteum, epiphyseal trabecular bone and within the bone marrow itself [250]. CGRP receptor expression was demonstrated in osteoblasts and bone marrow stromal cells, mediating the stimulatory effect of CGRP on osteoblast proliferation and differentiation [254,318,319]. Additionally, CGRP was identified to inhibit osteoclast differentiation and activity in vitro [320]. In bone regeneration, CGRP plasma levels were observed to be elevated in patients with long-bone fracture [321], similar to rabbits suffering from fracture with concomitant traumatic brain injury [322].

Substance P (SP) represents another sensory neuropeptide, which is released from sensory nerve endings. SP is synthesized from the tachykinin 1 (*TAC1*) gene and signals through neurokinin 1 receptors [323]. Physiologically, SP is known for its initiating role in neurogenic inflammation, which can be further potentiated by CGRP [323]. Following CNS injury, SP was reported to be substantially elevated [324] and associated with the promotion of brain edema, injury of neurons, crucial BBB disruption and subsequent poor functional outcome [325]. In bone, both osteoblasts and osteoclasts were shown to express neurokinin 1 receptors [318,326]. Although current evidence of SP on bone formation remains inconsistent [327], SP is proposed to stimulate osteoblast proliferation and differentiation in a dose-dependent manner while simultaneously promoting the formation and activity of osteoclasts [328]. In vivo, SP-deficiency was observed to cause a minor reduction in bone resorption but a major impairment of bone formation and mineralization [328]. In bone repair, SP is considered an important regulator of angiogenesis which promotes fracture healing [329], while the antagonization of its receptor neurokinin 1 results in reduced callus formation and biomechanical strength [330,331].

Different to the neuropeptides CGRP and SP, semaphorins represent a group of soluble and membrane-associated proteins, originally identified to regulate axonal guidance and growth [332]. Different semaphorins are considered to be involved in bone homeostasis, such as semaphorin 3A (sema3A), sema4D, sema6D and sema7A [333,334]. Osteoblast-derived sema3A, particularly important for sensory neuronal development, was discovered to feature significant osteoprotective properties by promoting osteoblast function and suppressing osteoclastic activity [335]. While studying conditional sem3A-deficient mice, only the depletion of sema3A in sensory neurons resulted in a lower BMD phenotype but not the inactivation of osteoblastic sema3A [336]. These results contradict direct regulatory effects of sema3A on osteoblasts, indicating an indirect modulation of bone remodeling through sensory nerve development [1,2,4,334,335,337].

In summary, these observations highlight the importance of the nervous system in bone homeostasis and regeneration. While SNS activity results in a suppression of bone formation, PSNS and SeNS activity are proposed to counterbalance this effect by promoting bone formation and reducing bone resorption [1,2,3]. The neuronal regulation of bone mass by SeNS-derived peptides exhibit exceptional features with notable future therapeutic potential.

### 3.2. Bone-Derived Mediators Acting on the Central Nervous System

Recent studies revealed a bidirectional dependence of brain and bone through bone cell-derived modulators that directly affect behavioral and cognitive function. The main bone-derived mediator affecting the brain represents osteocalcin (OCN), which is encoded by the *BGLAP* gene and exclusively synthesized by osteoblasts. Undercarboxylated, bioactive OCN, initially considered as an inhibitor of bone mineralization [337], participates in systemic body regulation and homeostasis [338] primarily regulating insulin secretion, muscle adaptation as well as testosterone production in the testis [6]. In addition, OCN was recently discovered to transverse the BBB to enter the CNS, where it promotes spatial learning and memory while preventing anxiety-like behavior or even depression [6]. Cognitive function and circulating levels of OCN are proposed to inversely correlate with age while maternal osteocalcin regulates embryonic brain development by enhancing monoamine neurotransmitters and their synthesis [339]. Although the treatment capacity of OCN in neurodegenerative disorders remains to be elucidated [340], models of movement disorders such as PD demonstrated a neuroprotective effect of OCN [341], while OCN was reported to improve insulin sensitivity and muscle strength [342]. Interestingly, the hormonal role of osteocalcin was currently challenged by two individually performed experimental studies [343,344] that generated novel murine osteocalcin-knockout strains which lacked the previously postulated endocrine dysregulation. Therefore, both OCN-deficient mice strains with a *Bglap*/*Bglap2* double-knockout allele displayed regular bone quantity, glucose metabolism, fertility [343] and muscle mass [344]. Bone microstructure analyses of the two studies revealed an increase in cortical bone carbonate-to-phosphate ratio and collagen maturity [343] but also the disruption of the biological apatite crystallographic orientation which resulted in reduced bone strength [344]. The impact of OCN-deficiency on cognition and brain development within the new models remains to be elucidated.

Another bone-derived mediator, lipocalin 2 (LCN2), also known as neutrophil gelatinase-associated lipocalin, is a glycoprotein which regulates energy metabolism by mediating insulin secretion and improving glucose tolerance as well as insulin sensitivity [345]. Similar to osteocalcin, the predominantly osteoblast-derived LCN2 was recently discovered to cross the BBB in order to activate the anorexigenic pathway by binding to the melanocortin 4 receptor (MC4R) in the hypothalamus [346]. Therefore, the suppression of appetite by LCN2 is referred to as an endocrine function of bone [346].

Along with the osteoblast-derived OCN and LCN2, the osteocyte-specific sclerostin was additionally identified to affect brain function. Sclerostin, a glycoprotein encoded by the *SOST* gene, antagonizes Wnt signaling by binding to the Lrp4/5/6 [347], which interrupts the Wnt/β-catenin pathway resulting in elevated bone resorption and reduced bone formation [348]. Therefore, the inhibition of sclerostin was proposed as a therapeutic approach in the treatment of bone loss [349], which led to the development and approval in the US and Europe of the monoclonal sclerostin-antibody Romosozumab for severe osteoporosis [350,351,352,353]. In the brain, Wnt/β-catenin signaling is essential for neurogenesis, neuronal survival, synaptic plasticity, BBB integrity [82] and further associated with the pathophysiology of AD [354]. As *SOST*/sclerostin is expressed by several other tissues in physiological and pathogenic conditions [355], additional studies are required to elucidate whether pharmacologic inhibition of sclerostin may also affect Wnt/β-catenin signaling in the brain [8].

Similar to sclerostin, dickkopf-related protein 1 (DKK1) represents another antagonist of Wnt signaling. In bone, DKK1 binds to Lrp6 also antagonizing the Wnt/β-catenin pathway [356], whereby over-expression of DKK1 results in osteopenia [357]. During early embryogenesis, DKK1 is released from the endoderm in order to control cell death and differentiation [358]. As a result, over-expression of DKK1 leads to synaptic loss and neuronal apoptosis, making it a common marker for neuronal death in neurodegenerative diseases [359]. In the aging brain, Wnt signaling was identified to generally diminish [360], while DKK1 remained up-regulated [361]. Such an elevation of DKK1 was clinically and experimentally reported for individuals suffering from AD, further supporting the hypothesis that Wnt signaling dysfunction contributes to the pathology of AD [362]. Along with its positive effect on bone, restoring the Wnt/β-catenin signaling is regarded as a promising future therapeutic approach to treat neurodegenerative diseases [354,359,362,363,364,365].

In addition to the molecular mediators secreted by bone, bone marrow-derived cells were identified to enter the systemic circulation and migrate into the injured brain [8]. In preclinical studies, bone marrow-derived microglia-like cells were shown to modulate amyloid pathology by restricting Aβ plaque formation and by supporting Aβ plaque clearance, which improved cognitive impairment [366,367,368]. Therefore, recent attempts employed hematopoietic stem cells mobilized from bone marrow into the peripheral blood for autologous microglia-like cell preparation [369], using specific antibodies to convert bone marrow cells into trafficking microglia-like cells [367]. Additionally, the systemic transplantation of bone marrow-derived macrophages reduced neuroinflammation with a limited reversion of Aβ deposition [370]. In summary, growing evidence of experimental AD studies demonstrates the capacity of bone marrow-derived microglia and macrophages to enter the CNS and to suppress the progression of brain degeneration, which provides a promising therapeutic tool for AD patients [8,371].

### 3.3. Mediators Affecting Both Brain and Bone Function

Both brain and bone share common regulatory factors which are locally synthesized in both organ systems. These mediators include irisin, osteopontin (OPN) and RANKL as well as the growth factors brain-derived neurotrophic factor (BDNF), IGF-1 and bone morphogenic proteins (BMPs). Irisin, a cleaved product of the precursor type 1 membrane protein, is described as an exercise-induced myokine [372,373] with the capacity to change visceral adipose tissue into brown adipose tissue, highlighting its potential role in preventing obesity or even metabolic syndrome [374,375]. In the brain, irisin was discovered to feature neuroprotective effects following ischemic stroke [376] through the activation of phagocytotic cells [377], while rescuing synaptic plasticity and memory defects in models of AD [375,377,378]. In bone, irisin showed osteoanabolic effects by enhancing osteoblastic activity and reducing the number of osteoclasts [372]. Furthermore, irisin was proposed to upregulate *RUNX2* and Wnt/β-catenin signaling, potentially transmitted by the suppression of the Wnt antagonist sclerostin [373]. As a result, irisin represents a notable pharmacologic target currently under discussion in osteoporosis-associated neurodegenerative diseases and metabolic research [7,374].

OPN represents a glycoprotein encoded by the *SPP1* gene, which is expressed in a variety of tissues including brain and bone. OPN is one of the non-collagenous proteins present in bone matrix, supporting bone demineralization by anchoring osteoclasts to bone mineral matrix which enhances the process of bone resorption [379,380]. Therefore, patients with high serum OPN levels are associated with low BMD [381]. Besides bone tissue, OPN is a multifunctional molecule with high expression in inflammatory diseases, where it serves as a proinflammatory cytokine [382]. In the brain, OPN is proposed to protect neurons and to regulate repair processes in various brain disorders such as ischemia, stroke, TBI and neurodegenerative diseases [77,383,384,385].

RANKL is encoded by the *TNFSF11* gene, also expressed in various tissues including brain and bone. In the CNS, RANKL/RANK are particularly present in the lateral septal nucleus of the hypothalamus, controlling the central regulation of fever and gender specific body temperature in females [386], whereas in bone, RANKL represents the key mediator for activation, differentiation, fusion and survival of osteoclasts [387]. For therapeutic treatment of bone loss diseases, the RANKL-neutralizing monoclonal antibody Denosumab [387,388] was developed, which is fully approved in the US and Europe. Recently, Denosumab was reported to improve chronic social defeat stress in mice [389,390], which is proposed as novel treatment indication in patients suffering from depression.

The growth factor BDNF is secreted in the CNS and PNS. In the brain, BDNF supports neurodevelopment, synaptic plasticity, neuronal differentiation and survival while its reduced levels are associated with neuronal dysfunction and degeneration in disorders such as depression, AD and PD [391]. Interestingly, the central inactivation of BDNF expression leads to gender-dependent increased bone mass [392], obesity and leptin resistance [393].

IGF-1, a polypeptide hormone structurally similar to insulin, is primarily synthesized in the liver in response to growth hormone signaling [394]. In the brain, IGF-1 was discovered to perform pleiotropic actions during development, alterations of neuronal excitability and enhancement of nerve cell metabolism, posing anti-apoptotic properties [395]. In bone, the growth hormone/IGF-1 axis stimulates growth directly by activating chondrocyte proliferation and osteoblast differentiation [396,397], while the reduction of IGF-1 serum levels and insulin-deficiency of type 1 diabetes were associated with osteoporosis [397].

Expressed in most body tissues, BMPs represent the largest subgroup of the transforming growth factor beta (TGFβ) family, regulating various developmental processes [398]. In the brain, BMPs are involved in neurogenesis with differential expression levels in adults [399,400], indicating their participation in age-related neuronal dysfunction [401]. Therefore, BMPs are discussed for therapeutic protection of the white matter [402]. In bone, BMPs enhance endochondral ossification and skeletal regeneration [400]. Interestingly, statins, commonly known for their application in cardiovascular diseases, were shown to promote osteogenesis via the BMP pathway by inhibiting osteoclastogenesis and apoptosis of osteoblasts [403,404,405], which highlights their osteoanabolic and anti-resorptive treatment capacity [406,407].

## 4. Conclusions

Several clinical observations provide key evidence for a bidirectional communication between brain and bone tissue, which is strongly supported by experimental studies that unraveled the underlying mechanistic pathways and identified molecular mediators involved in this crosstalk. The majority of brain-bone crosstalk is ’efferent,’ as the nervous system tightly modulates bone metabolism and regeneration. Thus, bone is regulated by multiple transmitters of both the CNS and PNS. Although evidence of the bone’s regulatory capacity is continuously growing, this ‘afferent’ loop has long been underestimated. Skeletal tissue emerged as a crucial modulator of cerebral development, function and pathologies. Investigating the diverse physiological and pathological interactions of brain and bone has thus revealed a variety of promising pharmacologic targets, supporting anabolic treatment strategies in bone and neuroprotective effects in brain pathologies. Future therapeutic approaches however need to consider the intense and dynamic brain-bone crosstalk as well as genetic and neuropsychological comorbidities of affected patients, potentially requiring additional monitoring or even individualized treatment regimes.

## Figures and Tables

**Figure 1 ijms-21-04946-f001:**
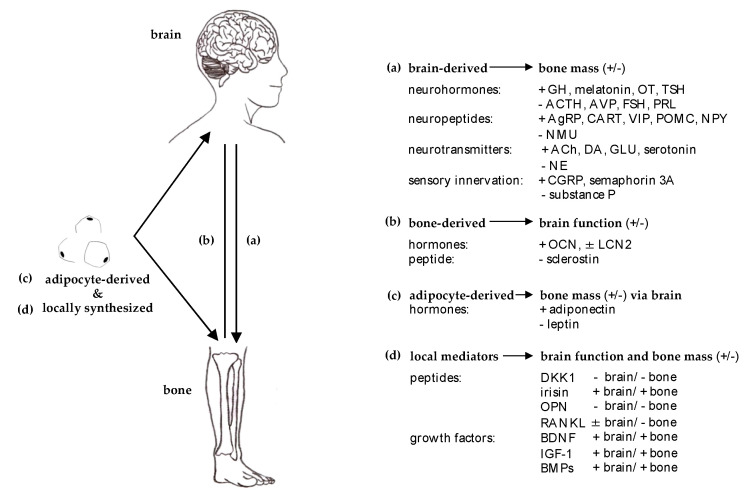
**Molecular bases of brain and bone crosstalk.** Summary of the predominant mediator effect of (**a**) brain-derived on bone, (**b**) bone-derived on brain, (**c**) adipocyte-derived on bone via central modulation and (**d**) locally synthesized mediators affecting brain and bone concurrently. The mediator effect on brain function and bone mass previously reported was summarized as positive (+) and negative (−). Abbreviations: ACh = acetylcholine, ACTH = adrenocorticotrophic hormone, AgRP = agouti-related peptide, AVP = arginine-vasopressin, BDNF = brain-derived neurotrophic factor, BMPs = bone morphogenic proteins, CART = cocaine amphetamine regulated transcript, CGRP = calcitonin gene related peptide, DA = dopamine, FSH = follicular stimulating hormone, DKK1 = dickkopf-related protein 1, GH = growth hormone, GLU = glutamate, IGF-1 = insulin-like growth factor 1, LCN2 = lipocalin 2, NE = norepinephrine, NMU = neuromedin U, NPY = neuropeptide Y, OCN = osteocalcin, OPN = osteopontin, OT = oxytocin, PRL = prolactin, POMC = proopiomelanocortin, RANKL = receptor activator of nuclear factor-κB ligand, TSH = thyroid-stimulating hormone, VIP = vasoactive intestinal peptide.

**Table 1 ijms-21-04946-t001:** Crosstalk of brain and bone: clinical observations.

	Origin	Condition	Effect on Bone	Clinical Studies/Reviews
**brain**	*neuro-psychological causes*	chronic stress and shift work	- higher fracture risk - increased proinflammatory cytokines, which stimulate bone resorption - shift workers have a higher risk for osteoporosis and fracture potentially caused by hormonal changes - melatonin has bone protective effects and improves sleep parameters	[13,14,25,41,42,44,45,157,158] reviewed by [159]
		major depressive disorder (MDD)	- low BMD and a higher risk for fracture - hypothalamic-pituitary-adrenal (HPA) axis dysregulation with higher levels of glucocorticoids, catecholamines and lower levels of steroids	[49,50,51,52,53,54,55,56,160]
		stroke	- reduced bone mineral density (BMD) - high serum concentration of bone turnover markers are found from early on - higher risk of heterotopic ossification (HO) - higher risk of fracture - bone fracture may affect ischemic stroke recovery	[16,62,63,64,65,66,67,68,69,161,162]
		dementia/Alzheimer’s disease (AD)	- lower BMD and increased fracture risk - less brain atrophy correlates with higher BMD - elevated osteopontin correlates with cognitive decline - AD progression linked to sclerostin, osteopontin/-calcin - abnormal Wnt/β-catenin signaling causes BBB dysfunction - Aβ plaques significantly enlarged in brain and bone, enhancing osteoclasts function	[8,73,74,75,76,77,78,86] reviewed by [163]
		Parkinson’s disease (PD)	- increases fracture risk - lower BMD in early stages	[23,87,88]
	*trauma*	traumatic brain injury (TBI)	- reduction of BMD after TBI - TBI frequently associated with HO - TBI with concomitant fracture showed an accelerated fracture healing and enlarged callus formation - beneficial effect of TBI restricted to closed fractures - increased osteogenic effects following TBI (serum/CSF-mediated)	[19,116,121,128,129,130,131,132,133,134,139,140] [164,165]
		spinal cord injury (SCI)	- isolated SCI associated with lower BMD and osteoporosis - SCI frequently associated with HO - SCI associated with accelerated fracture healing and enhanced callus formation in multitrauma	[20,145,146,147,148,149] [164] reviewed by [166]
			**Effect on Brain**	
**bone**	*chronic* *disorders*	osteoporosis	- associated with TBI, SCI, AD, PD, stroke and epilepsy - bidirectional impact of MDD, caused by fractures and pain leading to impaired quality of life	see each condition/disorder for reference; reviewed by [15]
	*genetic*	cleidocranial dysplasia (CCD)	- skeletal disorder characterized by skeletal anomalies - **brain**: developmental delay or late-onset progressive cognitive decline, suggested to be osteocalcin induced	reviewed by [90] [91,92]
		hereditary multiple exostoses (HME)	- skeletal disorder characterized by multiple osteocartilaginous overgrowths (exostoses), skeletal deformities - **brain**: nerve entrapment, chronic pain and association with autism or mental impairment e.g., dementia	[101,102,103]
		sclerosteosis and van Buchem disease	- skeletal disorder (autosomal recessive) characterized by generalized hyperostosis with bone anomalies - **brain**: mechanical impact causing raised intracranial pressure and entrapment of cranial nerves	[99,100] [167]
	*trauma*	fracture (Fx)	- increased Fx risk in neurodegenerative diseases and stroke, associated with reduced BMD - Fx history seen as independent risk factor of dementia - Fx patients at particular risk of concomitant mild TBI - peripheral injury caused higher functional deficits and mortality rates in patients suffering from TBI	[23] [168,169] reviewed by [22]
			**Effect on Brain and Bone**	
**brain and bone**	*genetic*	Coffin-Lowry syndrome (CLS)	- loss-of function mutations of gene encoding for RSK2 - **brain**: severe psychomotor retardation - **bone**: growth retardation with skeletal malformations	[94]
	*trauma*	complex regional pain syndrome (CRPS)	- systemic chronic pain condition after trauma/surgery, causing autonomic, sensory and motor abnormalities - **brain**: neurogenic inflammation, maladaptive neuro-plasticity and nociceptive sensitization with sensory gain/loss; potentially resulting in anxiety/depression - **bone**: loss of BMD and increased periarticular bone metabolism with OPG as a potential biomarker	reviewed by [21] [104,105,106,107]

**Table 2 ijms-21-04946-t002:** Crosstalk of brain and bone: experimental observations.

	Origin	Condition	Effect on Bone	Model	Pre-Clinical Studies/Reviews
**brain**	*neuro-psychological causes*	chronic stress and shift work	- proinflammatory cytokines stimulate bone resorption - glucocorticoids are released, directly inhibiting bone formation - disturbed fracture healing - the bone tissues circadian clock genes (*Bmal1*) enhances BMD and osteoblasts activity while decreasing osteoclastogenesis - melatonin with combined calcium carbonate improves osteoporosis (bone quality)	in vitro mouse mouse mouse/rat mouse	[43] [46,47] [48] [37,289,408] [409]
		depression	- low osteoblast differentiation	rat	[57]
		stroke	- fracture exacerbates ischemic cerebral injury	mouse	[71]
		dementia/Alzheimer’s disease (AD)	- Aβ increases osteoclastic activation - Wnt/b-catenin signaling ameliorate BBB function - dysfunctional Wnt/b-catenin signaling in AD	in vitro in vitro	[81,410] [411,412] reviewed by [163]
		Parkinson’s disease (PD)	- dopaminergic degeneration accelerates bone loss	mouse	[89]
	*trauma*	traumatic brain injury (TBI)	- reduction of BMD following TBI - TBI enhances the formation of heterotopic ossification (HO) - TBI with concomitant fracture showed an accelerated fracture healing and enlarged callus formation, potentially caused by: - dominance of neuronal mechanisms and neuroinflammation - calvaria anabolic response mediated by cannabinoid-1 receptor - leptin-deficiency eliminates positive effect - hippocampus and calcitonin gene-related peptide (CGRP) - SDF-1 promotes endochondral bone repair - elevated levels of leptin in CSF and GH/IGF-1 in serum - elevated serum arachidonic acid following TBI, which promotes the expression of BGLAP and therefore osteoblasts proliferation - elevated serum CGRP following TBI - elevated secretion of CGRP following TBI - close association of serum leptin and callus volume - release of osteogenic factors into the serum following TBI	mouse/rat rat mouse mouse mouse mouse rat mouse rabbit rat rat rat rat rat	[117,118] [126] [141,142] [144] [413] [414] [415] [416] [417] [418] [419] [420] [421] [422]
		spinal cord injury (SCI)	- SCI serum accelerates fracture healing	in vitro	[423]
			**Effect on Brain**		
**bone**	*chronic* *disorders*	osteoporosis	- associated with AD, potentially caused by the disruption of the Wnt/β-catenin signaling pathway	mouse	[424]
	*genetic*	cleidocranial dysplasia (CCD)	- absents of Runx2 showed bone resorption defect with reduced levels of osteoblast-produced osteocalcin in serum and brain - reduced brain osteocalcin was associated with enhanced anxious behavior and impaired cognitive function (as observed for CCD)	mouse	[425]
		hereditary multiple exostoses (HME)	- inactivation of *Ext1* resulted in exostoses as well as stereotypies and impairment of socio-communication, suggesting that mutated genes of heparan sulfate biosynthetic enzymes (incl. *EXT1*) are the cause for associated autism and other mental disorders	mouse mouse	[426] [427]
	*trauma*	fracture (Fx)	- long-bone fracture exacerbates TBI and neuroinflammation, with a worsened cerebral edema and neurological recovery	mouse	[155,156,428]
			**Effect on Brain and Bone**		
**brain and bone**	*genetic*	Coffin-Lowry syndrome (CLS)	- **brain**: deletion of (the in CLS) mutated RSK2 is associated with impaired spatial learning and long-term spatial memory deficit - **bone**: ATF4, substrate of *RSK2*, regulates osteoblast differentiation and activity, potentially contributing to skeletal phenotype of CLS	mouse mouse	[95] reviewed by [94] [97]

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
