# Peer review of "Crosstalk of Brain and Bone—Clinical Observations and Their Molecular Bases"

_ijms, 2020, doi:10.3390/ijms21144946_

Round 1

Reviewer 1 Report

  1. The manuscript is written well and educational for readers, presenting important subject.
  2. It would strengthen the manuscript if authors cite, if any, additional previous studies in which genetically-engineered mice with abnormal bone phenotype also demonstrate CNS/brain dysfunction and/or degeneration in Table 1 and Table 2.
  3. The manuscript may be easier to follow if factors and hormones are bulletized in the text. 

Reviewer 2 Report

The manuscript by Otto et al. very nicely sums up the subject of crosstalk between brain and bone from the point of view of clinical and molecular bases in much detail. The figure is representative and the tables are informative. In the opinion of the reviewer the shown material is an accurate overview of the subject. However, there is a lack of strong evidences like the real biological significance of bone as a crucial afferent regulator of brain development. This might not be evidently missing but would give the manuscript a so much more readable style that it should be added, if possible. In addition, the resolution of the tables should be improved prior to publication.

Reviewer 3 Report

Poznan, Poland, 2020-06-13

Review and comments to the manuscript:”Crosstalk of brain and bone – clinical observations and their molecular bases”.  

 The manuscript is very interesting and practice, the way of description is also perfect.

This review summarizes observations reported for the crosstalk of brain and bone, followed by a detailed overview of their molecular bases.

Authors showed the main pathways of the brain-bone crosstalk remain ’efferent’, signaling from brain to bone, this review emphasizes the emergence of bone as a crucial ‘afferent’ regulator of cerebral development, function and pathophysiology and central regulators, transmitters of the sympathetic, parasympathetic and sensory nervous system, bone-derived mediators altering brain function are released from bone cells and the bone marrow.

Presented manuscript will be suitable for publication after minor revision:

Comment:

                I suggest showing in the figure 1. and briefly in the text that the individual signal paths are controlled by individual genes and their polymorphisms. The authors mention the TNFSF11 gene, the CNR2 gene, the SPP1 gene, so they should write about the BMP2 gene, the FGFβ gene, the LRP5 gene and  at the discretion of the coding genes other factors listed on line: 259-266 (please include the most important. For the sake of order, I only show information about genetic conditioning, please complete it.

255: Figure 1. Molecular bases of brain and bone crosstalk.

259-266: ACh = acetylcholine, ACTH = adrenocorticotrophic hormone, AgRP= agouti-related peptide, AVP = arginine-vasopressin, BDNF = brain-derived neurotrophic factor, BMPs = bone morphogenic proteins, CART = cocaine amphetamine regulated transcript, CGRP = calcitonin gene related peptide, DA = dopamine, FSH = follicular stimulating hormone, DKK1 = dickkopf-related protein 1, GH = growth hormone, GLU = glutamate, IGF-1 = insulin-like growth factor 1, LCN2 = lipocalin 2, NE = norepinephrine, NMU = neuromedin U, NPY = neuropeptide Y, OCN = osteocalcin, OPN = osteopontin, OT = oxytocin, PRL = prolactin, POMC = proopiomelanocortin, RANKL = receptor activator of nuclear factor-κB ligand, TSH = thyroid- stimulating hormone, VIP = vasoactive intestinal peptide.

417-419: Gut-derived serotonin was reported to decrease osteoblast proliferation in a low-density lipoprotein receptor-related protein 5 (Lrp5)-dependent manner [253], as well as to promote maturation of osteoclasts resulting in elevated bone resorption [254].

636-638: OPN represents a glycoprotein encoded by the SPP1 gene, which is expressed in a variety of tissues including brain and bone. OPN is one of the non-collagenous proteins present in bone matrix, supporting bone demineralization by anchoring osteoclasts to bone mineral matrix which enhances the process of bone resorption [371,372].

644-647: RANKL is encoded by the TNFSF11 gene, also expressed in various tissues including brain and bone. In the CNS, RANKL/RANK are particularly present in the lateral septal nucleus of the hypothalamus, controlling the central regulation of fever and gender specific body temperature in females [378], whereas in bone, RANKL represents the key mediator for activation, differentiation, fusion and survival of osteoclasts [379].

664: Expressed in most body tissues, BMPs represent the largest subgroup of the transforming growth factor beta (TGFβ) family, regulating various developmental processes [392].

465: In clinical studies, genetic variants of the CNR2 gene, encoding for CB2 receptors, were associated with low BMD and osteoporosis [293].

               I would like to emphasize that the article is a very good, innovative summary of the signal pathways connecting the brain to the bone, based on rich literature. It is read with real pleasure, congratulations to the authors.
